# Comparison between Colistin Sulfate Dry Powder and Solution for Pulmonary Delivery

**DOI:** 10.3390/pharmaceutics12060557

**Published:** 2020-06-17

**Authors:** Frédéric Tewes, Julien Brillault, Nicolas Gregoire, Jean-Christophe Olivier, Isabelle Lamarche, Christophe Adier, Anne-Marie Healy, Sandrine Marchand

**Affiliations:** 1INSERM U1070 “Pharmacology of anti-infective agents”, 1 rue Georges Bonnet, Pôle Biologie Santé, 86022 Poitiers Cedex, France; julien.brillault@univ-poitiers.fr (J.B.); nicolas.gregoire@univ-poitiers.fr (N.G.); Isabelle.Lamarche@univ-poitiers.fr (I.L.); christophe.adier@inserm.fr (C.A.); sandrine.marchand@univ-poitiers.fr (S.M.); 2UFR Médecine-Pharmacie Université de Poitiers, 6 rue de la milétrie, TSA 51115, 86073 Poitiers Cedex 9, France; jean.christophe.olivier@univ-poitiers.fr; 3Laboratoire de Toxicologie-Pharmacocinétique, CHU de Poitiers, 2 rue de la Miletrie, 86021 Poitiers Cedex, France; 4SSPC, The Science Foundation Ireland Research Centre for Pharmaceuticals, School of Pharmacy and Pharmaceutical Sciences, Trinity College Dublin, Panoz Institute, D02 PN40 Dublin 2, Ireland; healyam@tcd.ie

**Keywords:** colistin dry powder inhalation, pulmonary infections, nebulization, pharmacokinetics modeling, Calu-3 permeability

## Abstract

To assess the difference in the fate of the antibiotic colistin (COLI) after its pulmonary delivery as a powder or a solution, we developed a COLI powder and evaluated the COLI pharmacokinetic properties in rats after pulmonary administration of the powder or the solution. The amorphous COLI powder prepared by spray drying was characterized by a mass median aerodynamic diameter and fine particle fraction of 2.68 ± 0.07 µm and 59.5 ± 5.4%, respectively, when emitted from a Handihaler^®^. After intratracheal administration, the average pulmonary epithelial lining fluid (ELF): plasma area under the concentration versus time curves (AUC) ratios were 570 and 95 for the COLI solution and powder, respectively. However, the same COLI plasma concentration profiles were obtained with the two formulations. According to our pharmacokinetic model, this difference in ELF COLI concentration could be due to faster systemic absorption of COLI after the powder inhalation than for the solution. In addition, the COLI apparent permeability (P_app_) across a Calu-3 epithelium model increased 10-fold when its concentration changed from 100 to 4000 mg/L. Based on this last result, we propose that the difference observed in vivo between the COLI solution and powder could be due to a high local ELF COLI concentration being obtained at the site where the dry particles impact the lung. This high local COLI concentration can lead to a local increase in COLI P_app_, which is associated with a high concentration gradient and could produce a high local transfer of COLI across the epithelium and a consequent increase in the overall absorption rate of COLI.

## 1. Introduction

Due to its low permeability across the lung–blood barrier, colistin methanesulfonate (CMS) is one of only four antibiotics currently used to treat *Pseudomonas aeruginosa* lung infections by inhalation. CMS is an anionic prodrug of colistin (COLI), which is a cationic lipopeptide mix composed mainly of two molecules: colistin A and B. Inhaled COLI therapy is almost always conducted by means of nebulization of CMS instead of the cationic COLI since the use of the latter can lead to more side effects, such as bronchoconstriction, throat irritation, and cough [1]. However, CMS is not an optimal prodrug. In fact, CMS is inactive against *P. aeruginosa* and has to be converted by non-enzymatic hydrolysis into COLI to produce its bactericidal effect [2]. This conversion is slow compared to the CMS pulmonary absorption rate [3,4,5,6]. In addition, in critically ill patients receiving CMS by nebulization, only 1.4% (*w*/*w*) of the CMS dose is converted into COLI in the lung [4]. As a consequence, relatively high CMS concentrations are required to obtain effective COLI concentrations in the pulmonary epithelial lining fluid (ELF) [3,4,5]. Additionally, it was shown that CMS conversion into COLI is highly variable between patients [4], which could lead to inefficient or toxic COLI concentrations in the ELF after CMS nebulization.

Consequently, the development of cationic COLI-based formulations for inhalation has recently regained attention [7,8,9,10]. For antibiotics, the aim of pulmonary inhalation is to administer large doses in a short time frame and in a reproducible manner. This can be obtained by using either dry powder inhalers (DPIs) or solution nebulizers. In a previous study, the pulmonary function of patients and volunteers was evaluated after inhalation of pure COLI sulfate salt powder simply blended with lactose powder [11]. In some patients, a moderate to severe cough was observed. However, the authors suggested that this drawback might be overcome by an improvement in the formulation design. In addition, one of the aims of the study was to develop a dry powder formulation for inhalation, which could produce a pulmonary residence time for COLI, which is equivalent to that obtained with a solution. There are many differences between inhaled antibiotic therapy delivered by DPIs and by nebulizers [12]. In particular, these two methods of inhalation might produce different local COLI concentrations and distributions that could lead to different pharmacokinetic (PK) profiles and efficacy. For that reason, we developed and characterized an inhalable COLI-based dry powder made by spray drying a solution of COLI and inulin and evaluated its PK properties in rats in comparison with a COLI aqueous solution.

## 2. Materials and Methods

Colistin sulfate salt (COLI) was purchased from Sigma–Aldrich (St. Quentin Fallavier, France). Inulin (INU) with a degree of polymerization of 8 fructosyl moieties was kindly provided by Sensus (Roosendaal, The Netherlands). The chemicals used in this study were of analytical grade, and the solvents were of high-performance liquid chromatography (HPLC) grade.

### 2.1. Pseudoternary Phase Diagram

Water–methanol–butyl acetate solvent mixture was used as the solvent for preparing the spray-dried solutions [13,14]. The combined proportions of the 3 solvents, which resulted in clear solutions in the presence of the solutes used to formulate the particles, were determined by pseudoternary phase diagrams. The diagrams were constructed using, as solute, either INU (0.75% *w*/*w*) or INU (0.5% *w*/*w*) and COLI (0.5% *w*/*w*). Multiple tubes containing 0.5 mL of total volume were prepared with solvents volume ratio changing with a 10% step. The resulting mixtures were allowed to reach equilibrium for 1 h while stirring at 25 °C. The macroscopic aspect of the tubes was visually analyzed to determine the number of phases and the turbidity of the phases.

### 2.2. Spray Drying

Water–methanol–butyl acetate solutions containing only INU at a concentration of 0.75% (*w*/*w*) or a 1/1 (*w*/*w*) mixtures of INU and COLI having a total concentration of 1% (*w*/*w*) were spray-dried using a B-290 Mini spray dryer (Büchi, Flawil, Switzerland) set in the closed cycle mode with a 2-fluid nozzle having a 0.7 mm diameter hole and the nozzle cap of 1.4 mm diameter. Spray drying parameters were set as for previously optimized porous microparticle production (of raffinose and trehalose), as previously described [15]. The inlet temperature was 100 °C, and the feeding pump was set at 30% (8 mL/min). Nitrogen flowing at 15 L/min was used as the spraying gas. Nitrogen flowing at 630 L/min (aspirator rate 100%) was used as the drying gas in a co-current mode. These conditions resulted in an outlet temperature ranging from 52 to 55 °C and were previously found to produce saccharide powders with less residual methanol and butyl acetate than the concentration limits allowed by the European Pharmacopoeia [15].

### 2.3. Characterization of the Particles

#### 2.3.1. Scanning Electron Microscopy (SEM)

Pictures of powders were taken using a Mira XMU (Tescan, Brno, Czech Republic) SEM. The dry powders were coated with a 10 nm-thick gold film and scanned by electrons accelerated under a voltage of 5 kV. Images were formed from the collection of secondary electrons.

#### 2.3.2. Particle Size Distribution Analysis

The volume-based particle size distributions (PSD) were determined by laser diffraction using a Mastersizer 2000 (Malvern Instruments Ltd., Worcestershire, UK) with the Scirocco 2000 dry powder feeder to disperse the particles. The dispersive air pressure used was 2 bar. Samples were run at a vibration feed rate of 50%. Analyses were undertaken using a real part refractive index of 1.520 and an absorption part of 0.1. Results presented are the average of three determinations

#### 2.3.3. Aerodynamic Particle Diameter Analysis

The aerodynamic diameter (AD) distribution of the particles was measured using a Next Generation Impactor (NGI, Copley Scientific Limited, Nottingham, UK). The Handihaler^®^ (Boehringer Ingelheim) higher-resistance inhaler was filled with a size 3 hard gelatin capsule loaded with 20 ± 2 mg of powder (n = 3). The flow rate was adjusted to 50 L/min to obtain 4 kPa of pressure drop across the inhaler and time duration of aspiration was set to obtain 4 L. After the capsules contents were released, the masses of particle remaining in the capsule, the inhaler, the mouthpiece adapter, and deposited in the induction port and all the stages of the NGI were determined, after powder dissolution in an appropriate volume of water by the COLI assay described below. The number of particles with AD ≤ 5.0 µm, expressed as a percentage of the emitted recovered dose, was considered as the fine particle fraction (FPF). The mass median aerodynamic diameter (MMAD) and FPF were calculated as previously described [13,16]. All the experiments were performed in triplicate.

#### 2.3.4. Powder X-Ray Diffraction (XRD)

XRD measurements were conducted on samples placed in a low background silicon holder, using a Rigaku Miniflex II desktop X-ray diffractometer (Rigaku, Tokyo, Japan). The samples were scanned over a range of 5 to 40° 2*θ* at a step size of 0.05°/s.

### 2.4. Pharmacokinetics

#### 2.4.1. Animals

In vivo experiments were undertaken in compliance with EC Directive 2010/63/EU after agreement by a local ethics committee. Male Sprague–Dawley rats (n = 68) from Janvier Laboratories (Le Genest-St.-Isle, France) with weights of 324 ± 24 g (mean ± SD) were used. Animals were acclimatized for 5 days prior to experiments. They always had free access to food and water.

#### 2.4.2. Intratracheal Administration of COLI as Particles or Solution

The target COLI dose delivered was 0.35 mg kg^−1^ (base form). This dose was previously used in our lab to investigate the pharmacokinetic properties of colistin following intratracheal administration of colistin sulfate solution in rats [17]. Pulmonary administrations were performed after a sedation period with isoflurane (3% air at 550 mL min^−1^ for 10 min). For COLI solution nebulization, 100 µL of INU-COLI solution at a COLI concentration of 1.25 mg mL^−1^ in saline was spray instilled using the PennCentury MicroSprayer™ aerosolizer (model IA–1B, Penn-Century, Philadelphia, PA, USA). For INU-COLI powder, intratracheal insufflation, around 0.3 mg of powder (0.15 mg of COLI) was delivered in one puff of 2 mL of air using a Dry Powder Insufflator–Model DP-4 (Penn-Century, Philadelphia, PA, USA). The insufflator was weighed with a 0.01 mg accuracy scale (Sartorius KB BA100) before and after insufflation to determine the exact amount of powder delivered, and the dose of colistin administered was calculated based on the animal’s weight.

#### 2.4.3. Samples for the Plasma PK Study (n = 44)

The day before the experiment polyethylene catheters were implanted into the femoral artery of anesthetized rats. Previous studies showed that plasma COLI half-life was 30–40 min. Accordingly, arterial blood samples were collected at 0, 0.25, 0.5, 1, 2, 3, and 4 h after pulmonary administration [18]. Plasma was obtained and frozen at −80 °C.

#### 2.4.4. Samples for Determination of Concentrations in ELF (n = 24)

Bronchoalveolar lavage fluid (BALF) collection was performed according to methods previously described [6,17]. Briefly, at 0.5, 2, and 3 h after COLI administration, 1 mL of NaCl (0.9%) at 37 °C was instilled into the trachea of anesthetized rats (4 to 6 per group) via a catheter (50-mm depth). BALFs were immediately collected, and then blood was obtained via heart puncture. COLI and urea concentrations were determined in both plasma and BALF. Since urea diffuses rapidly in tissues, its concentration is the same in plasma as in ELF. Therefore, the COLI concentration in ELF(C_ELF_) was calculated using Equation 1 from measured COLI concentrations in BALF (C_BAL_) by using urea as a marker of dilution [17].
C_ELF_ = C_BAL_ (Urea_plasma_/Urea_BAL_)(1)
where Urea_BAL_ and Urea_plasma_ correspond to the concentrations of urea determined in BALF and plasma.

#### 2.4.5. Simultaneous PK Modeling of Plasma and ELF Concentrations of COLI

Plasma and ELF COLI concentrations versus time were simultaneously analyzed by a nonlinear mixed-effects method with S-ADAPT software (v 1.52, Biomedical Simulations Resource (BMSR), Los Angeles, USA) with S-ADAPT TRAN translator [19]. The structural model (Appendix C, Figure A3) used for data analysis has been developed in a previous study to fit plasma and ELF COLI concentrations after COLI intravenous administrations and solution nebulization to rats [17]. Briefly, the systemic pharmacokinetics of COLI were described by a two-compartment model, characterized by a central compartment (V_c_), a total systemic clearance (CL_T_), and a peripheral compartment (V_p_) connected by an equilibrium distribution clearance (Q). The lung part of the model consisted of a depot compartment with a first-order transfer rate (k_a_), a systemic bioavailability parameter (F_neb_), and an ELF compartment with a fixed physiological volume (V_ELF_) estimated at 30 µL·kg^−1^. The distribution between the ELF compartment and the central systemic compartment consisted of a two-way diffusion clearance (Q_ELF_) and a nonlinear influx transfer (from ELF to the central compartment) implemented with a transfer rate (V_in_) characterized by Equation (2).
(2)$$V_{\mathrm{in}}=\frac{V_{\max,\mathrm{in}}\;.\;C^{\gamma }}{k_{m}{}^{\gamma }{+C}^{\gamma }}$$
where V_max,in_ is the maximum influx transfer rate, k_m_ corresponds to the concentration for which V_in_ = 0.5V_max,in_ and γ is the slope factor. Only the unbound drug was assumed to distribute between plasma and lung compartments. COLI unbound plasma fraction was fixed at 45% [20]. The residual variability was estimated with a proportional error model in plasma and ELF. Plasma concentrations below the limit of quantification were handled by the Beal M3 method [21]. Statistical comparison of parameter values between COLI administered as a solution, dry powder, or dissolved particles was carried out by likelihood ratio tests with a significance threshold for type I error of 5%. When values were not significantly different between formulations, a common parameter value was estimated. Typical systemic parameters (V_c_, CL_T_, V_p_, Q) were taken to be identical whatever the formulation administered via the pulmonary route. Typical values and inter-individual variability of PK parameters are reported (Appendix D, Table A1) along with the precisions of the estimates expressed as relative standard errors (RSE). Areas under plasma concentration and ELF concentration time-curves (AUC)_0→∞_ were calculated by numerical integration of the model predictions with Berkeley Madonna software (version 8.3.18; University of California).

### 2.5. In-Vitro Transport Study

#### 2.5.1. Calu-3 Cell Culture

Calu-3 cells were purchased from the American Type Culture Collection (Manassas, VA, USA). The cells were cultured in DMEM/Ham’s F12 (1/1) (PanBiotech) supplemented with l-glutamine (2 mM), fetal calf serum (10% *v*/*v*), and incubated at 37 °C under 90–95% of RH and 5% *v/v* of CO_2_ in the air.

#### 2.5.2. Transport Study

Transport experiments were conducted in apical-to-basolateral directions to determine COLI apparent permeability (P_app_) as previously described [6]. Briefly, the Calu-3 cells at passages 40–60 were seeded at a density of 5 × 10^5^ cells/cm^2^ onto 12-well plate Transwell inserts. The cells were cultured under air-interface conditions for 15 days. On the day of the experiment, Calu-3 monolayers were rinsed with transport medium made of Hanks’ Balanced Salt solution (HBSS) buffered at pH 7.4 with 10 mM HEPES. Following a 30 min equilibration period, the transport medium in the donor compartment (apical side) was replaced by fresh transport medium containing COLI or the solubilized INU-COLI particle at a concentration of 100 mg/L to 4000 mg/L of COLI, and incubated for 6 hrs. Then, sample aliquots were taken from the acceptor compartment (basolateral side). Following the transport study, a control of the monolayer integrity was performed by measuring sodium fluorescein P_app_ [22]. Briefly, the monolayers were rinsed and fresh transport medium was added in the basolateral side, and a solution of sodium fluorescein in the transport medium (10 µg/mL) was added in the apical side. Samples were taken after 60 min at 37 °C. P_app_ value of 0.7 × 10^−6^ cm·s^−1^ for fluorescein was selected as the value describing good tight junction integrity [23].

### 2.6. Analytical Assays

#### 2.6.1. HPLC Assay of Colistin

This method was applied to quantify COLI in spray-dried powder and in samples collected from the NGI impactor. The chromatography was performed with an X-Terra MS C18 column (5.0 µm, 150 mm, 3.9-mm i.d.; Waters, St.-Quentin en Yvelines, France) and a mobile phase consisting of acetonitrile and an aqueous buffer composed of sodium sulfate (4.5 g/L) and orthophosphoric acid (4.5 g/L) (23:77, *v*/*v*) flowing at 0.5 mL/min. Eluted COLI was detected using a UV detector set at 210 nm. The calibration standard curve was prepared by injecting (25 µL) six samples in water with COLI base concentrations ranging between 30 and 500 µg·mL^−1^.

#### 2.6.2. COLI Analysis in Plasma, BALF, and Calu-3 Transport Medium

The determination of COLI plasma concentrations was performed based on a validated LC-MS/MS method previously described [17,18,24]. The chromatography was performed on an Alliance 2695 system (Waters, France) with a Jupiter 300-Å column (5.0 µm, 50 mm, 2.0-mm i.d.; Waters, St.-Quentin en Yvelines, France) and a mobile phase (flow rate 0.2 mL/min) consisting of 0.1% (*v*/*v*) formic acid in acetonitrile–0.1% formic acid in water (25:75 *v*/*v*). The mass spectrometer Micromass Quattro microAPI (Waters, France) was used in the positive/ion mode. The calibration standard curve was prepared with seven samples in rat plasma with COLI base concentrations ranging between 0.0097 and 10 µg·mL^−1^. For analysis of COLI in BALF and in the medium used for the COLI transport experiments, 150 µL of sample was mixed with 100 µL of plasma and analyzed as for plasma.

#### 2.6.3. Urea Analysis in BALF and Plasma

BALFs urea concentrations were determined by LC-MS/MS, as previously described [6,17]. Briefly, standard curves were prepared in NaCl (0.9% *m*/*v*) with urea concentrations ranging from 1.25 to 100 µg·mL^−1^. The column used was an Xterra MS C18 column (5.0 µm, 150 by 4.6-mm ID; Waters, St. Quentin en Yvelines, France) and the mobile phase (flow rate 0.25 mL·min^−1^) was made of 0.1% (*v*/*v*) formic acid in acetonitrile–0.1% formic acid in water (10:90 *v*/*v*). The concentrations of urea in plasma were measured using a cobas^®^ modular automatic analyzer (Meylan, Roche Diagnostics, France).

## 3. Results and Discussion

### 3.1. Microparticle Formulation and Characterization

Colistin sulfate dry powder for inhalation has recently regained attention [7,8,9,10]. In a past study, the pulmonary function of patients and volunteers was evaluated after inhalation of pure COLI sulfate blended with lactose dry powder [11]. In some patients, a moderate to severe cough was observed. However, the authors suggested that this drawback might be overcome by an improvement in the formulation design. In the present study, COLI-loaded particles for pulmonary inhalation were developed by using inulin (INU) as an excipient allowing particles with suitable aerodynamic properties to be formulated [25,26] and potentially reducing the irritating effects observed in humans with pure COLI sulfate. Additionally, INU was shown to be an excellent stabilizing agent for amorphous material due to its high glass transition temperature [27]. Besides, INU was previously shown to improve the stability of deoxyribonuclease dry powder that was suitable for inhalation and produced by spray-drying [26].

A water–methanol–butyl acetate (H_2_O-MeOH-BA) solvent mixture was used to prepare the solutions to be spray-dried. Spray drying from this solvent system was previously shown to result in porous particles of saccharides that had good dispersion characteristics, making them suitable for inhalation [13,15]. The first step of the study was to determine the proportions of the three solvents, which resulted in clear solutions of the particle ingredients (solutes). Accordingly, pseudo-ternary phase diagrams were constructed from the three solvents in the presence of INU or INU and COLI (Figure 1). In the presence of INU alone, three areas were identified in the phase diagram (Figure 1A). One of the areas was constituted of one clear phase, which contains a maximum of 20% of BA and a minimum of 50% of MeOH. A second area, composed of two clear phases, was obtained for BA concentrations higher than 10% and MeOH concentrations lower than 40%. A third area was composed of a cloudy phase, resulting from the precipitation of INU. This area was obtained by an increase in BA concentration. In the presence of 0.5 *w*/*w*% INU and 0.5 *w*/*w*% COLI, the phase diagram was also composed of three areas but with different boundary limits (Figure 1B). With COLI and INU, the cloudy phase was extended towards higher MeOH concentrations compared to INU alone. The pseudo-ternary phase diagrams showed that only a small area made of H_2_O:MeOH:BA in a volume proportion of 30:50:20 was suitable to produce a clear solution. This proportion was then used to prepare the feed solutions for the spray dryer.

The boiling temperatures of MeOH and H_2_O are lower than that of BA. Thus, the sprayed droplets become concentrated in BA during the spray-drying process, resulting in the precipitation of the solutes (cloudy state) as suggested by the arrows in the phase diagram (Figure 1). Particles formed by spray-drying with an early phase separation process occurring during the drying phase show a surface accumulation due to the high Peclet number of the precipitate [28]. The resulting particles are hollow spheres if the shell becomes quickly rigid, or are of a wrinkled or dimpled morphology if the surface buckles or folds on the internal void space [28]. In both cases, particles generally have a low apparent density, and dimpled particles also have a low interparticle contact area. These two characteristics are advantageous for a DPI formulation [29].

Spray-dried particles made of pure INU or made of INU and COLI in a 1:1 weight ratio had different morphologies (Figure 2). INU particles were spherical with a smooth surface, but INU-COLI particles were dimpled. The irregular dimpled surface should lead to low interparticulate contacts, which generally results in decreased cohesive interactions between particles and in readily dispersible powders [29].

The geometric particle size distribution (PSD) measured by laser diffraction was slightly shifted to larger sizes in the presence of COLI compared to particles of INU alone (Figure 3A). D_(50)_ values of the PSD of INU and INU-COLI particles were 1.86 ± 0.24 µm and 2.42 ± 0.31 µm, respectively. Aerodynamic properties of the INU-COLI particles (Figure 3B) demonstrated suitability for powder pulmonary inhalation, having a mass median aerodynamic diameter (MMAD) of 2.68 ± 0.07 µm and a geometric standard deviation (GSD) of 1.98 ± 0.04. This MMAD value is in the range of the breathable fraction of an aerosol, with an MMAD between 1 and 5 μm, considered suitable for topical respiratory treatment [29]. Particles having an MMAD between 1 and 3 μm have high peripheral deposition compared with particles having an MMAD close to 5 μm [30]. The fine particle fraction (FPF), defined as the percentage of inhaled fine particle mass (particles having an aerodynamic diameter less than 5 µm) relative to the total mass of particles emitted from the DPI, was 59.5 ± 5.4%.

The solid-state nature of the spray-dried powder was determined by powder XRD (Appendix A
Figure A1). XRD patterns of raw and spray-dried powders presented only diffuse halos, characteristic of XRD amorphous materials. After a dynamic vapor sorption (DVS) cycle, the INU-COLI formulation was found to remain XRD amorphous (Appendix B
Figure A2), indicating good physical stability for the INU-COLI particles.

### 3.2. Pharmacokinetics

Plasma and pulmonary ELF COLI concentrations versus time profiles measured and predicted by the PK model (described in Appendix C
Figure A3) after intratracheal administration of COLI solution or INU-COLI powders are presented in Figure 4. After pulmonary administration of COLI as INU-COLI powder (Figure 4B), or as solutions made by dissolving either the COLI raw material powder (Figure 4A) or the INU-COLI spray-dried powder (Figure 4C), similar COLI plasma profiles were obtained.

The peak of COLI concentration in plasma occurred at 0.5 h after administration and the area under the concentration versus time curves (AUC) were comparable. Thus non-significantly different typical bioavailabilities of 64% (RSE = 6%) were obtained after pulmonary administration of INU-COLI powder, INU-COLI solution, or COLI solution (Appendix D
Table A1). This observation is in accordance with the results published by Lin et al. [10], where they described similar systemic bioavailabilities (~46.5%, RSE = 8.43%) measured in rats after intratracheal delivery of pure COLI solution and pure COLI powder at a dose of 0.66 mg/kg. After pulmonary administration, COLI concentrations were much higher in ELF than in plasma, whatever the formulation used. The variability in ELF COLI concentrations measured after the administration of COLI solutions seemed similar to those obtained after the administration of CMS solution in rats, using the same method of spray instillation [6]. However, ELF COLI concentration values were more variable after intratracheal administration of the INU-COLI powder than the COLI solution.

The presence of INU did not influence the PK of COLI after pulmonary administration. In fact, the ELF:plasma AUC typical ratios were both 570 after the nebulization of the COLI solutions made either from the COLI raw material or from the INU-COLI spray-dried powder (Figure 4A,C). However, this ratio was decreased to 95 after the pulmonary administration of the INU-COLI dry powder (Figure 4B). Thus, the typical COLI AUC in ELF was 6-fold lower after the intratracheal delivery of INU-COLI powders compared to solutions at the same dose.

Pulmonary exposure to COLI after administration of aerosol depends on the dose that reaches the lung and on various clearance mechanisms, such as mucociliary clearance, macrophage phagocytosis, and systemic absorption [31]. COLI has been reported to have an effect on the integrity of membranes and has been shown to be able to increase the transport of molecules through them, acting as a permeation enhancer [32,33]. Thus, the effect of the COLI concentration on its apparent permeability (P_app_) across a Calu-3 epithelium model was evaluated to estimate its possible influence on its systemic absorption after pulmonary administration.

### 3.3. In-Vitro Transport Study on Calu-3 Broncho-Epithelial Cell Line

COLI P_app_ changed as a function of COLI concentration in the Calu-3 cell monolayer apical side (Figure 5). For a COLI concentration of 100 mg/L, a P_app_ of 0.43 ± 0.11 × 10^−7^ cm·s^−1^ was determined. This value was consistent with a previous determination (0.42 × 10^−7^ cm·s^−1^) [6]. COLI P_app_ measured for the same COLI concentration obtained by dissolving INU-COLI particles was similar (0.55 ± 0.12 × 10^−7^ cm·s^−1^), showing that INU did not influence the transport of COLI through the epithelium. At this concentration, COLI had a relatively low P_app_, suggesting that a low in-vivo absorption rate of COLI should occur after its intratracheal delivery. These data are consistent with the PK results observed in rats after COLI solution pulmonary administration [17] or in humans after CMS solution pulmonary nebulization [4].

An increase in COLI concentration to 1000 mg/L induced a 4-fold increase in COLI P_app_. A further increase in COLI concentration to 4000 mg/L induced an additional 2.5-fold increase in COLI P_app_. In this latter condition, the P_app_ of fluorescein across the Calu-3 monolayers measured after the COLI transport study was increased by 2.7-fold compared to the control measurements carried out only with the transport medium (Appendix E
Table A2 and Table A3). This increase in fluorescein P_app_ suggested an alteration of the monolayer integrity. It is important to highlight that these last two COLI concentrations are more than 100-times higher than the average COLI ELF concentration required to have maximal efficacy. However, Boisson et al. [4] showed that the COLI concentrations measured in the ELF of patients after the nebulization of CMS solutions could be greater than 1000 mg/L six hours after inhalation. In addition, due to the low volume of ELF in the rat (approximately 30 µL/kg—i.e., 10 µL per rat) [17] compared to the volume administered (100 µL), the concentration of COLI obtained in the ELF of the rat after nebulization of the colistin solution must be close to the concentration of the solution administered (1250 mg/L), which is in the middle of the concentration range tested on the Calu-3 cells.

The increase in COLI P_app_ observed with its increase in concentration is in accordance with previously reported studies. Lewis et al. [32] showed that the transepithelial conductance of the rabbit bladder epithelium increased with the COLI concentration. Likewise, the increase in liposome permeability induced by COLI was shown to be concentration-dependent [33]. COLI interacts with membranes primarily via the insertion of the alkyl chains or through the formation of mixed micelles [34]. Our results suggest that these interactions also modified the Calu-3 cells membrane permeability to COLI. One of the aims of pulmonary drug delivery is to reach high local concentration while reducing the systemic concentrations that are toxic to the kidney. Therefore, COLI pulmonary relative absorption rate may increase with the COLI ELF concentration due to the increase in its P_app_, challenging the benefits of reaching a very high concentration by pulmonary delivery.

What are the mechanisms that could explain a lower COLI AUC in pulmonary ELF after administration of COLI dry powder compared to the AUC measured after administration of COLI solution? The AUC of colistin in pulmonary ELF depends on the dose fraction that has reached the ELF and on its clearance from the ELF (through systemic absorption, mucociliary clearance, macrophage clearance). A lower COLI AUC in pulmonary ELF obtained with COLI particles could be due to their significant mucociliary clearance or to their phagocytosis by pulmonary macrophages, while these mechanisms should not have an impact on the COLI solution. However, in that case, we should also have observed a lower COLI plasma AUC after the powder administration than after the administration of the solutions. Still, similar COLI plasma AUCs were obtained after solution or powder pulmonary delivery.

Consequently, the only reason remaining that can explain a difference in COLI AUC in pulmonary ELF is a difference in systemic absorption. Indeed, according to our PK model, the difference in COLI ELF concentrations obtained after the delivery of the COLI powder or solutions is due to a more rapid systemic absorption of COLI with the powder compared to the solutions. The faster absorption was characterized by a Km value, i.e., the COLI concentration in the ELF for which the transfer rate from the ELF compartment to the central compartment is Vmax/2, which was significantly (six times) lower for INU-COLI particles than for COLI solutions (Appendix D
Table A1). This result is in agreement with the work of Lin et al. [10], who also found a higher COLI rate of absorption after pure COLI powder intratracheal delivery than after the delivery of COLI as a solution. In their study, Lin et al. [10] observed a reduction in plasma Tmax measured after administration of the COLI dry powder (10 min—first sampling time) compared to that obtained after administration of the COLI solution (20 to 30 min). In the present study, Tmax and plasma Cmax were equal to 0.5 h and 0.33 mg/L, respectively, whatever the formulation used, so the difference in systemic absorption rate had little impact on these parameters. The PK model predicts that during the first half-hour post-administration (when the COLI concentrations in ELF were greater than 10 mg/L), the factor limiting the passage into the plasma was the rate of transfer from the deposit compartment, which was identical for both formulations. Therefore, the increase in the absorption rate of COLI with the dry powder occurs roughly at the same time as Tmax, leading to similar Tmax and Cmax values for the INU-COLI dry powder and COLI solution.

Two mechanisms can potentially explain faster systemic absorption of COLI after pulmonary administration of COLI in the form of solid particles compared to a COLI solution. One hypothesis is that the small and amorphous particles of INU-COLI dissolve quickly in the pulmonary ELF. Amorphous INU dry particles loaded with a lipophilic drug (THC) and prepared in comparable conditions had a fast dissolution [27]. This fast dissolution should result in high local concentrations of COLI in the ELF at the point where they impacted the pulmonary epithelium. According to studies performed on the Calu-3 epithelium model, this high local COLI ELF concentrations could result in an increase in the COLI P_app_ locally where the particles landed. Thus, a high local concentration associated with a high local P_app_ should create high local COLI fluxes through the pulmonary epithelium, which could explain the overall increase in the absorption rate of COLI. When COLI was administered as a solution, the maximal COLI concentration reachable in the ELF was equal to the COLI concentration in this solution (1250 mg·L^−1^). This limited the increase in COLI P_app_ and the concentration gradient of COLI through the pulmonary epithelium. A similar mechanism was previously proposed to explain the better efficiency of absorption enhancers incorporated in inhaled dry powders compare to their administration as solutions [35,36]. A second hypothesis is that the INU-COLI dry powder particles and COLI solution droplets could deposit in different lung areas of different absorption capacities. Several studies have shown that the alveolar region has a higher absorption capacity than the upper airways [37]. Thus a faster absorption rate could be explained by a higher peripheral deposition of the INU-COLI particles compared to the deposition obtained with the COLI solution. This hypothesis was the one selected by Lin et al. [10] to explain the faster COLI systemic absorption after administration of their dry powder compared to a solution of COLI. However, several studies that have evaluated the area of exposure produced by the MicroSprayer IA-1B^®^ (Penn–Century, Philadelphia, PA, USA) in rats and mice by scintigraphic imaging acquired after intratracheal delivery of molecules labeled with 99mTc have shown a good peripheral exposure [38,39].

In pharmacokinetic/pharmacodynamic studies of COLI systemically administered to model mice of Pseudomonas aeruginosa lung infection, Cheah et al. [40] found that the target values of fAUC/MIC for COLI in plasma should be much higher than 10, even higher than 100 for certain strains, for COLI to be effective. After IV administration of COLI, AUCs in plasma and in pulmonary ELF are comparable. This suggests that exposure to colistin in the lungs should be much higher than in plasma to be effective. Therefore, from a clinical point of view, the results of the present study suggest that the use of a quick-dissolve/release COLI-based powder is of less interest than a solution of COLI. However, the development of a controlled release formulation would allow better control of the COLI concentration in the pulmonary ELF and would increase its residence time in the lungs as well as its efficacy.

## 4. Conclusions

This study comparing pulmonary administration of colistin sulfate as a dry powder or solution showed lower exposure of the pulmonary epithelial lining fluid to colistin after administration of the powder compared to the solution. The exposure of pulmonary epithelial lining fluid to colistin depends on the dose fraction that has reached the fluid and the rate of elimination from that fluid. (through systemic absorption, mucociliary clearance, macrophage clearance). Similar plasma exposure to colistin was obtained with liquid and powder formulations, eliminating the involvement of mucociliary and macrophage clearance mechanisms and suggesting a faster systemic absorption rate of colistin after the powder intratracheal insufflation than after the solution spray nebulization. This difference in the absorption rate could be explained by the higher local colistin concentration obtained when particles of a dry powder land on the epithelial lining fluid, creating a high concentration gradient across the blood–lung barrier and increasing the colistin apparent epithelium permeability as observed on a Calu-3 epithelium model.

## Figures and Tables

**Figure 1 pharmaceutics-12-00557-f001:**
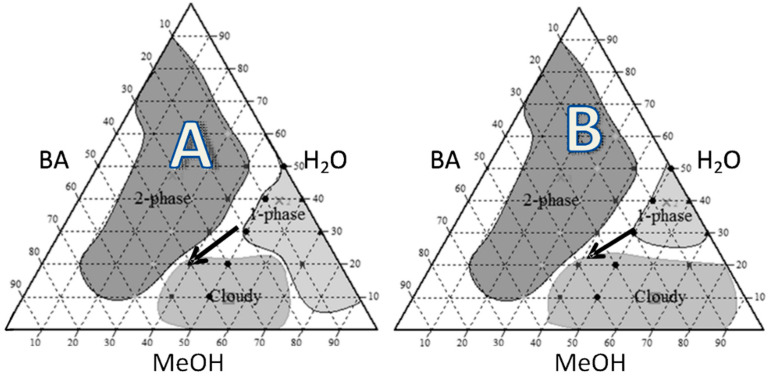
Butyl acetate–methanol–Water pseudoternary phase diagrams obtained in the presence of (**A**) inulin (INU) at 0.75 *w*/*w*% and, (**B**) a mix of INU at 0.5 *w*/*w*% and colistin (COLI) 0.5 *w*/*w*%. The arrow in the diagram indicates the relative change in solvent composition that should occur during the spray drying.

**Figure 2 pharmaceutics-12-00557-f002:**
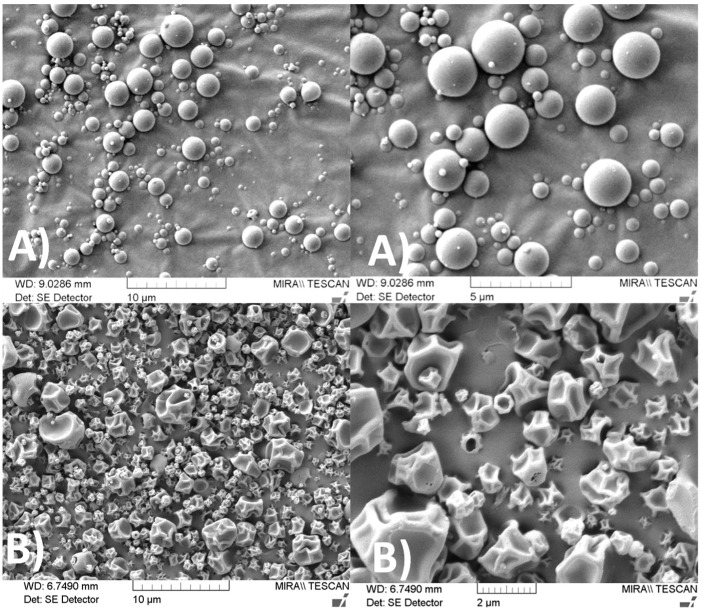
SEM micrographs of (**A**) spray-dried INU, (**B**) INU-COLI. The samples were coated with a 10 nm-thick gold film. Primary electrons were accelerated under a voltage of 5 kV. Images were formed from the collection of secondary electrons.

**Figure 3 pharmaceutics-12-00557-f003:**
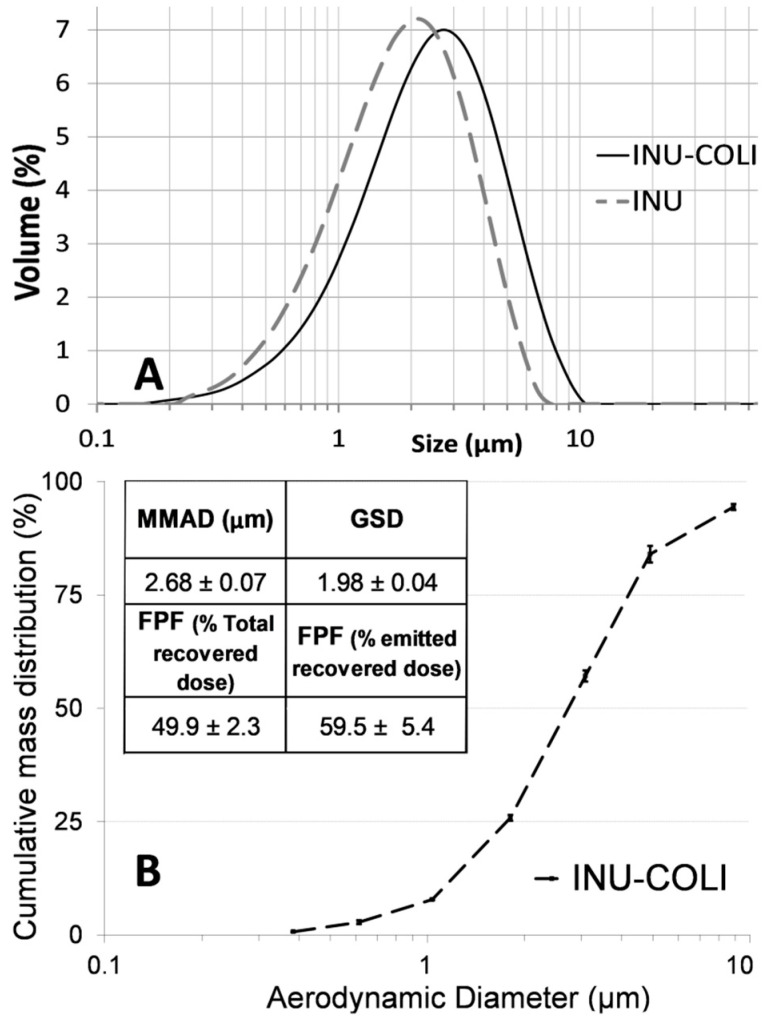
(**A**) Geometric particle size distribution determined by laser diffraction (n = 3). (**B**) Cumulative fraction of the mass of INU-COLI collected on the Next Generation Impactor (NGI) stages versus the effective cut-off diameter of the stages (n = 3).

**Figure 4 pharmaceutics-12-00557-f004:**
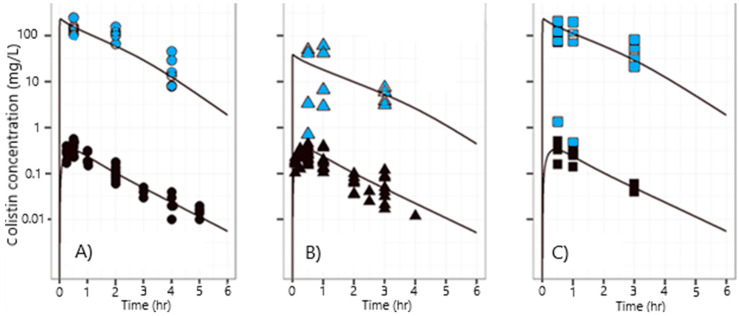
COLI concentrations measured in plasma (black symbols) and epithelial lining fluid (ELF) (blue symbols) and typical prediction of the model (solid lines) after intratracheal administration of (**A**) COLI solution as published by Gontijo et al. [17]; (**B**) INU-COLI dry powder; (**C**) COLI solution made by dissolving the INU-COLI powder. COLI dose delivered was 0.35 mg·kg^−1^ (base form).

**Figure 5 pharmaceutics-12-00557-f005:**
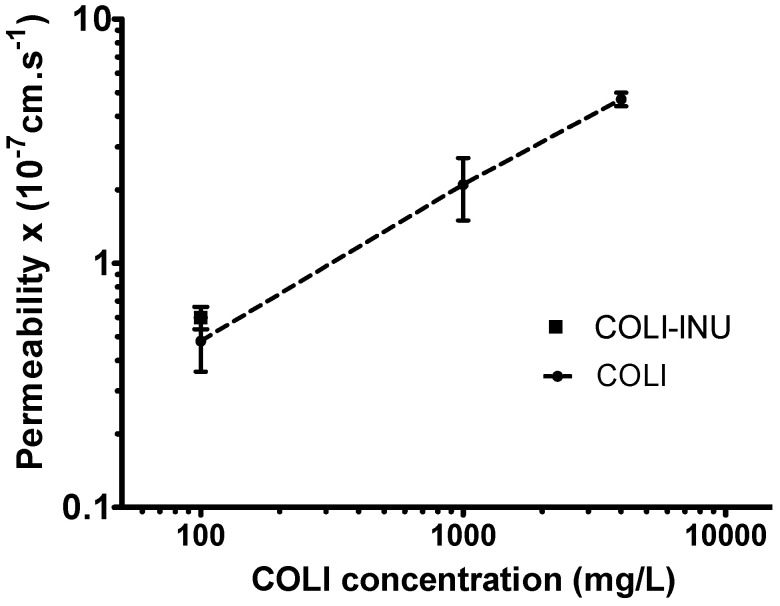
Apical-to-basal COLI apparent permeability across a Calu-3 cell epithelium model measured after 6 h of incubation of the apical side monolayer in the presence of different COLI concentrations (100, 1000, and 4000 mg/L for pure COLI and 100 mg/L for INU-COLI particle).

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
