# Peer review of "Comparison between Colistin Sulfate Dry Powder and Solution for Pulmonary Delivery"

_pharmaceutics, 2020, doi:10.3390/pharmaceutics12060557_

Round 1

Reviewer 1 Report

This manuscript intended to develop and characterize an inhalable colistin sulfate (COLI) dry powders by spray drying a solution of COLI and inulin with a particular view to evaluating the pharmacokinetic properties in rats in comparison with a COLI aqueous solution. The work is interesting and well performed and the manuscript is carefully prepared with a clear structure. Overall, this manuscript is of scientific significance and can be acceptable for publication.

Comments: The major finding of this study was that the pulmonary administration of colistin sulfate as a dry powder resulted in a faster systemic absorption rate than the solution based counterpart. Nonetheless, the Cmax and Tmax, PK parameters indicative of the rate of absorption, were not present and the data shown in Figure 4 appeared not to exhibit an increased Cmax in plasma, please explain in more details in the revised manuscript.

Author Response

We thank the reviewer for this valuable comment and have now included the following section in the discussion to clarify this point:

“Indeed, according to our PK model, the difference in COLI ELF concentrations obtained after the delivery of the COLI powder or solutions is due to a more rapid systemic absorption of COLI with the powder compared to the solutions. The faster absorption was characterized by a Km value, i.e. the COLI concentration in the ELF for which the transfer rate from the ELF compartment to the central compartment is Vmax/2, which was significantly (6 times) lower for INU-COLI particles than for COLI solutions (Appendix D – Table A1). This result is in agreement with the work of Lin et al. [10], who also found a higher COLI rate of absorption after pure COLI powder intratracheal delivery than after the delivery of COLI as a solution. In their study, Lin et al. [10] observed a reduction in plasma Tmax measured after administration of the COLI dry powder (10 min - first sampling time) compared to that obtained after administration of the COLI solution (20 to 30 min). In the present study, Tmax and plasma Cmax were equal to 0.5 h and 0.33 mg/L whatever the formulation used, so the difference in systemic absorption rate had little impact on these parameters. The PK model predicts that during the first half hour post-administration (when the COLI concentrations in ELF were greater than 10 mg/L), the factor limiting the passage into the plasma was the rate of transfer from the deposit compartment, which was identical for both formulations. Therefore, the increase in the absorption rate of COLI with the dry powder occurs roughly at the same time as Tmax, leading to similar Tmax and Cmax values for the INU-COLI dry powder and COLI solution.

This time needed to increase colistin absorption rate could correspond to the time needed for colistin to increase the permeability of the membrane

Reviewer 2 Report

The manuscript is generally clear, concise, well presented and written. The study itself is overall scientifically sound and although its originality is questionable, it will be worthy of publication following some improvements and clarifications.

  • It is not obvious whether the authors expected the powder formulation to improve colistin PK and/or retention in the ELF or on the other hand, their aim was to produce a more convenient powder equivalent of the colistin solution. A clinical perspective on the powder formulation developed and the PK data obtained should be included in the manuscript. 
  • Relatively harsh solvents were used during the spray-drying process. What about their residual content in the dry powder? Couldn't the solvents have caused some toxicity in the lungs or to the cells in vitro? Please, comment. 
  • Figure 4. It is quite blurry and therefore, its resolution would have to be increased before publication. 'Closed' and 'solid symbols' are very confusing. The authors might consider 'open' and 'solid' or 'clear' and 'dark' instead. Finally, the plasma conc profile for the solution (panel C) appears to include a lower number of data points (especially, the early time points) than the one for the dry powder (panel B). Why is this the case? Could the 'missing time points' result in artifically similar PK profiles between the powder and the solution? 
  • Page 9, lines 336-337: the sentence does not sound correct. Please, double check. 
  • Page 9, last sentence (lines 345-346). Do the authors imply colistin has permeation enhancer properties? Please, clarify that sentence.   
  • The range of colistin concentrations for the Calu-3 studies seemed to have been decided upon based on the ELF concentrations. However, the ELF covers the entire surface of the airways while Calu-3 layers have a surface of around 1 cm2. The in vitro concentrations would have been much higher than the concentrations that could have been reached locally on a surface of 1cm2 in vivo. Please, comment. 
  • Page 11, lines 365-366. That sentence is extremely vague. Furthermore, the fluorescein permeability data must be presented for each conditions and the 'control measurements' clearly defined. A major concern with the in vitro data is that the cells will have been stressed after 6-7h in HBSS/HEPES and without those data, the origin of the increase in Papp with the increase in colistin concentration is not clear. 
  • It is very difficult to understand why colistin absorption rate is higher for the powder than for the solution while their AUC are the same. The authors' preferred hypothesis is that the powder increases the permeability of the airway epithelium. This is not fully convincing as wouldn't that lead to a higher AUC as well? On the other hand, mucocilliary clearance might not affect the AUC if the drug is rapidly absorbed as it is the case for colistin. 
  • Considering the size of the droplets emitted by the Microsprayer is in the range 16-20 microns, it is very likely that the powder would have deposited deeper into the lung. That would have increased the absorption rate but also, potentially the likelyhood of clearance by alveolar macrophages. Do the ELF concentrations include colistin that might have been trapped within the cellular components of the BALF? The authors might want to comment on macrophage clearance. 

Author Response

Reviewer 2

The manuscript is generally clear, concise, well presented and written. The study itself is overall scientifically sound and although its originality is questionable, it will be worthy of publication following some improvements and clarifications.

  • It is not obvious whether the authors expected the powder formulation to improve colistin PK and/or retention in the ELF or on the other hand, their aim was to produce a more convenient powder equivalent of the colistin solution. A clinical perspective on the powder formulation developed and the PK data obtained should be included in the manuscript. 

The aim of the study was to compare the PK profiles of colistin after its administration as a dry powder or a solution, and produce a more convenient powder equivalent to the colistin solution. This information was added to the introduction of the revised manuscript.

A clinical perspective on the powder formulation was added to line 436 as follows: From a clinical point of view, the results of this study suggest that the use of a quick-dissolve/release COLI-based powder is of less interest than a solution of COLI. However, further development of a controlled release formulation would have the advantage of having a dry powder, which allows the administration of a dose in a few seconds, and would allow better control of the COLI ELF concentration and increase its lung residence time.

  • Relatively harsh solvents were used during the spray-drying process. What about their residual content in the dry powder? Couldn't the solvents have caused some toxicity in the lungs or to the cells in vitro? Please, comment. 

The European Pharmacopeia classifies methanol as a class 2 and butyl acetate as a class 3 residual solvent. In a previous study (Amaro, M.I.; Tewes, F.; Gobbo, O.; Tajber, L.; Corrigan, O.I.; Ehrhardt, C.; Healy, A.M. Formulation, stability and pharmacokinetics of sugar-based salmon calcitonin-loaded nanoporous/nanoparticulate microparticles (NPMPs) for inhalation. International journal of pharmaceutics 2015, 483, 6-18.), spray dried powders made of saccharides (raffinose and trehalose) using the same methanol/butyl acetate solvents blend and the same spray-drying inlet temperature showed residual methanol and butyl acetate content measured by gas chromatography lower than 0.3% (w/w) and 0.5 % (w/w), respectively, which are the concentration limits authorized by the European Pharmacopeia. This information was added to the revised manuscript in the section “2.2 Spray drying” as follows: “These conditions resulted in an outlet temperature ranging from 52 to 55 °C and were previously found to produce saccharide powders with less residual methanol and butyl acetate than the concentration limits allowed by the European Pharmacopoeia [15].

In addition, the PK profiles obtained for colistin after nebulization of colistin solutions made from commercial colistin or from the spray-dried INU-COLI powder were similar, suggesting that the compounds present in the powder did not influence the PK. Also, in vitro transport studies across the Calu-3 cell line were performed using the commercial colistin powder dissolved in the transport medium. Thus, the concentration-effect observed on the apparent permeability of colistin cannot be attributed to the effect of solvent.

  • Figure 4. It is quite blurry and therefore, its resolution would have to be increased before publication. 'Closed' and 'solid symbols' are very confusing. The authors might consider 'open' and 'solid' or 'clear' and 'dark' instead. Finally, the plasma conc profile for the solution (panel C) appears to include a lower number of data points (especially, the early time points) than the one for the dry powder (panel B). Why is this the case? Could the 'missing time points' result in artifically similar PK profiles between the powder and the solution? 

We have tried to improve the quality of the figure and add color to help distinguish the different conditions. The PK profiles in panel C of Figure 4 were obtained after intratracheal administration of a COLI solution made by dissolving the spray dried INU-COLI powder. This experiment was carried out to assess whether the inulin present in the powder could explain the difference observed between the COLI-INU powder (panel B) and the pure COLI solution (panel A). After obtaining the data for 3 different sampling times, we could see that no difference was observed between the 2 solutions (panel A and C). This is why did not need to perform more experiments with this condition.

  • Page 9, lines 336-337: the sentence does not sound correct. Please, double check. 

Thank you for noticing. The sentence, which is now line 395, has been modified as follows: “The faster absorption was characterized by a Km value, i.e. the COLI concentration in the ELF for which the transfer rate from the ELF compartment to the central compartment is Vmax/2, which was significantly (6 times) lower for INU-COLI particles than for COLI solutions”

  • Page 9, last sentence (lines 345-346). Do the authors imply colistin has permeation enhancer properties? Please, clarify that sentence.   

Yes, several studies have shown that colistin can modify the properties of biological barriers such as cell membranes or epithelium. We have clarified the sentence in the revised manuscript as follows: “COLI has been reported to have an effect on the integrity of membranes and has been shown to be able to increase the transport of molecules through them, acting as a permeation enhancer”

  • The range of colistin concentrations for the Calu-3 studies seemed to have been decided upon based on the ELF concentrations. However, the ELF covers the entire surface of the airways while Calu-3 layers have a surface of around 1 cm2. The in vitro concentrations would have been much higher than the concentrations that could have been reached locally on a surface of 1cm2in vivo. Please, comment. 

Due to the low volume of rat ELF (approximately 30 µL / kg – i.e. 10 µL per rat) compared to the administered volume (100 µL), the concentration obtained in the ELF of rats after nebulization of the colistin solution must be close to the concentration of the solution administered (1250 mg/L). This information was added at line 366 to the revised manuscript as follows “In addition, due to the low volume of ELF in the rat (approximately 30 µL/ kg – i.e. 10 µL per rat) [17]compared to the volume administered (100 µL), the concentration of COLI obtained in the ELF of the rat after nebulization of the colistin solution must be close to the concentration of the solution administered (1250 mg/L), which is in the middle of the concentration range tested on the Calu-3 cells..”

The small amorphous dry particles of INU-COLI should dissolve quickly in the ELF lung volume where they land, which could lead to local COLI concentrations at the point where they impact the pulmonary epithelium that are even higher than those obtained with the solution

The concentrations tested on the cells also correspond to the concentrations of colistin which can be found in humans after CMS nebulization. Indeed, Boisson et al. (Boisson, M.; Jacobs, M.; Grégoire, N.; Gobin, P.; Marchand, S.; Couet, W.; Mimoz, O. Comparison of intrapulmonary and systemic pharmacokinetics of colistin methanesulfonate (cms) and colistin after aerosol delivery and intravenous administration of cms in critically ill patients. Antimicrobial agents and chemotherapy 2014, 58, 7331-7339) have shown that the COLI concentrations measured in the ELF of patients after nebulization of the CMS solution are very variable and could be greater than 1000 mg/L six hours after inhalation.

  • Page 11, lines 365-366. That sentence is extremely vague. Furthermore, the fluorescein permeability data must be presented for each conditions and the 'control measurements' clearly defined. A major concern with the in vitro data is that the cells will have been stressed after 6-7h in HBSS/HEPES and without those data, the origin of the increase in Papp with the increase in colistin concentration is not clear. 

Thank you for noticing. The sentence on fluorescein permeability has been modified to be more precise as follows: In this latter condition, the Papp of fluorescein across the Calu-3 monolayers measured after the COLI transport study was increased by 2.7-fold compared to the control measurements carried out only with the transport medium (Appendix E – Table A2 and A3).

And the following table was added in the Appendix E – Table A2

Table A2: Apical-to-basal fluorescein permeability measured across a Calu-3 cell epithelium model after 6 hours of incubation of the apical side monolayer in the presence of different COLI concentrations

Colistin µg/ml

Fluorescein Papp (10-7 cm/s)

100

0.62 ± 0.08

1000

0.63 ± 0.18

4000

1.73 ± 0.02

The concentrations and incubation times for the Calu-3 experiments were chosen according to the analytical sensitivity of the LC-MS/MS method. Due to the very low permeability of colistin, a minimum concentration of 100 µg/ml in the apical cell compartment had to be used for transport experiments and a minimum time of 6 hours was needed in order to have concentrations of colistin above the limit of quantification in the basal receiver compartment. Concentrations lower than 100 µg/ml led to below-the-LOQ measurements unless the incubation time was increased to more than 6 hours. As noted by reviewer 2, this 6 hours incubation time is already a concern since the cells might by stressed in a HBSS/HEPES environment. However, we had undertaken experiments showing that HBSS incubation for 1, 3 or 6 hours did not significantly change the fluorescein permeability. Appendix E – Table 3

Incubation time in HBSS/HEPES (hrs)

Fluorescein Papp (10-7 cm/s)

1

0.54 ± 0.03

3

0.51 ± 0.04

6

0.64 ± 0.06

Thus, the increase in fluorescein Papp after 6 hours of colistin treatment is due to the presence of colistin at 4000 µg/ml.

  • It is very difficult to understand why colistin absorption rate is higher for the powder than for the solution while their AUC are the same. The authors' preferred hypothesis is that the powder increases the permeability of the airway epithelium. This is not fully convincing as wouldn't that lead to a higher AUC as well? On the other hand, mucocilliary clearance might not affect the AUC if the drug is rapidly absorbed as it is the case for colistin. 

The AUC reflects the extent of absorption of a drug, not the rate of its absorption. Colistin plasma AUC depends on the fraction of the dose that reached plasma and of the clearance from plasma, but does not depend on the rate at which colistin reaches the plasma. It can be assumed that the systemic clearance was comparable whatever the formulation (i.e. the formulation did not affect the ability of the rats to eliminate colistin from the plasma). Therefore, since the AUCs in plasma were comparable it can be assumed that the fraction of the dose that reached plasma was identical for every formulation (although rates of absorption into plasma may differ).

Likewise, the AUC of colistin in pulmonary ELF depends on the dose fraction that has reached the ELF and on its clearance from the ELF (systemic absorption, mucociliary clearance, macrophage clearance …). However, mucociliary clearance or phagocytosis by macrophages would have reduced the dose fraction that reached plasma. As the fraction of dose that reached the plasma was the same for all formulations, it is reasonable to assume that the same fraction of dose reached the ELF, and that mucociliary or macrophage clearances were negligible in these experimental conditions.

This point is now explained in the revised manuscript at line 382 as follows:

What are the mechanisms that could explain a lower COLI AUC in pulmonary ELF after administration of COLI dry powder compared to the AUC measured after administration of COLI solution? The AUC of colistin in pulmonary ELF depends on the dose fraction that has reached the ELF and on its clearance from the ELF (through systemic absorption, mucociliary clearance, macrophage clearance …). A lower COLI AUC in pulmonary ELF obtained with COLI particles could be due to their significant mucociliary clearance or to their phagocytosis by pulmonary macrophages, while these mechanisms should not have an impact on the COLI solution. However, in that case, we should also have observed a lower COLI plasma AUC after the powder administration than after administration of the solutions. Still, similar COLI plasma AUCs were obtained after solution or powder pulmonary delivery

  • Considering the size of the droplets emitted by the Microsprayer is in the range 16-20 microns, it is very likely that the powder would have deposited deeper into the lung. That would have increased the absorption rate but also, potentially the likelyhood of clearance by alveolar macrophages. Do the ELF concentrations include colistin that might have been trapped within the cellular components of the BALF? The authors might want to comment on macrophage clearance. 

The mucociliary clearance or phagocytosis by macrophages of the INU-COLI particles would have reduced the dose fraction that reached the plasma. As the fraction of dose that reached the plasma was the same for all formulations, it is reasonable to assume that mucociliary or macrophage clearances were negligible in these experimental conditions.

This point is now made in the revised manuscript at the line 382 as follows: “What are the mechanisms that could explain a lower COLI AUC in pulmonary ELF after administration of COLI dry powder compared to the AUC measured after administration of COLI solution? The AUC of colistin in pulmonary ELF depends on the dose fraction that has reached the ELF and on its clearance from the ELF (through systemic absorption, mucociliary clearance, macrophage clearance …). A lower COLI AUC in pulmonary ELF obtained with COLI particles could be due to their significant mucociliary clearance or to their phagocytosis by pulmonary macrophages, while these mechanisms should not have an impact on the COLI solution. However, in that case, we should also have observed a lower COLI plasma AUC after the powder administration than after administration of the solutions. Still, similar COLI plasma AUCs were obtained after solution or powder pulmonary delivery

Several studies that have evaluated the area of deposition produced by the MicroSprayer IA-1B® in rats and mice by scintigraphic imaging acquired after intratracheal delivery of molecules labeled with 99mTc have shown a good peripheral deposition.

 (Maillet, A.; Guilleminault, L.; Lemarié, E.; Lerondel, S.; Azzopardi, N.; Montharu, J.; Congy-Jolivet, N.; Reverdiau, P.; Legrain, B.; Parent, C. The airways, a novel route for delivering monoclonal antibodies to treat lung tumors. Pharmaceutical research 2011, 28, 2147-2156.

Gagnadoux, F.; Le Pape, A.; Urban, T.; Montharu, J.; Vecellio, L.; Dubus, J.-C.; Leblond, V.; Diot, P.; Grimbert, D.; Racineux, J.-L. Safety of pulmonary administration of gemcitabine in rats. Journal of aerosol medicine 2005, 18, 198-206.)

Recently, Wu et al. (L. Wu, C. Rodríguez-Rodríguez, D. Cun, M. Yang, K. Saatchi, U.O. Häfeli, Quantitative Comparison of Three Widely-Used Pulmonary Administration Methods In Vivo with Radiolabeled Inhalable Nanoparticles, European Journal of Pharmaceutics and Biopharmaceutics (2020), doi: https://doi.org/10.1016/j.ejpb.2020.05.004 ) also showed a good peripheral deposition of suspension droplets from intratracheal spraying with the Penn Century MicroSprayer IA-1B®

Round 2

Reviewer 2 Report

The authors have satisfactorily addressed all comments, provided the requested missing information and clarified previously confusing points. 

I do not have further comments. 

Author Response

The manuscript compared lung and plasmatic levels of colistin administered in solution or ias dry powder. It is well written in general . The content is not a great novelty but it can provide practical information. For that purpose, I miss any correlation between lung/plasma levels and differences if any in terms of antimicrobial activity, in order to support one of both types of formulations, if it is the case

The PK/PD targets are known in plasma but not in lung.  In pharmacokinetic / pharmacodynamic studies of COLI systemically administered to model mice of Pseudomonas aeruginosa lung infection, Cheah et al. found that the target values of fAUC/MIC for COLI in plasma should be much higher than 10, even higher than 100 for certain strains, for COLI to be effective. After IV administration of COLI, AUCs in plasma and in pulmonary ELF are comparable. This suggests that colistin concentrations in the lungs should be much higher than in plasma to be effective. Therefore, to treat lung infections, the higher the lung exposure to colistin, the better the effect (as long as toxic concentrations are not reached). Therefore, a better effect should be obtained using a solution than a quick-dissolving powder. Nevertheless, a dry powder formulation which provides a controlled release of COLI should allow better control of its concentration in ELF and increase its pulmonary residence time and efficiency. This point was added to the discussion in the manuscript on line 436 as follows.

In pharmacokinetic/pharmacodynamic studies of COLI systemically administered to model mice of Pseudomonas aeruginosa lung infection, Cheah et al.[42] found that the target values of fAUC/MIC for COLI in plasma should be much higher than 10, even higher than 100 for certain strains, for COLI to be effective. After IV administration of COLI, AUCs in plasma and in pulmonary ELF are comparable. This suggests that exposure to colistin in the lungs should be much higher than in plasma to be effective. Therefore, from a clinical point of view, the results of the present study suggest that the use of a quick-dissolve/release COLI-based powder is of less interest than a solution of COLI. However, the development of a controlled release formulation would allow better control of the COLI concentration in the pulmonary ELF and would increase its residence time in the lungs as well as its efficacy.”

Ref : Cheah S-E, Wang J, Nguyen VTT, Turnidge JD, Li J, Nation RL. New pharmacokinetic/pharmacodynamic studies of systemically administered colistin against Pseudomonas aeruginosa and Acinetobacter baumannii in mouse thigh and lung infection models: smaller response in lung infection. J Antimi-crob Chemother. 2015;70:3291–7